# A Literature Overview of Secondary Peritonitis Due to Carbapenem-Resistant Enterobacterales (CRE) in Intensive Care Unit (ICU) Patients

**DOI:** 10.3390/antibiotics11101347

**Published:** 2022-10-02

**Authors:** Sveva Di Franco, Aniello Alfieri, Marco Fiore, Ciro Fittipaldi, Vincenzo Pota, Francesco Coppolino, Pasquale Sansone, Maria Caterina Pace, Maria Beatrice Passavanti

**Affiliations:** 1Department of Women, Child and General and Specialized Surgery, University of Campania Luigi Vanvitelli, Piazza Miraglia 2, 80138 Naples, Italy; 2Department of Postoperative Intensive Care Unit and Hyperbaric Oxygen Therapy, A.O.R.N. Antonio Cardarelli, Viale Antonio Cardarelli 9, 80131 Naples, Italy; 3Unit of Critical Care Hospital “Ospedale Pellegrini”, Via Portamedina alla Pignasecca 41, 80134 Naples, Italy

**Keywords:** β-lactam/β-lactamase inhibitors, carbapenemase, combination therapy, critically ill patients, intra-abdominal infections, *Enterobacterales*, postoperative peritonitis, precision medicine, secondary peritonitis

## Abstract

This comprehensive review of the recently published literature offers an overview of a very topical and complex healthcare problem: secondary peritonitis from multidrug-resistant pathogens, especially carbapenem-resistant *Enterobacterales* (CRE). Spontaneous secondary peritonitis and postsurgical secondary peritonitis are among the major causes of community- and healthcare- acquired sepsis, respectively. A large number of patients enter ICUs with a diagnosis of secondary peritonitis, and a high number of them reveal infection by CRE, *P. aeruginosa* or *A. baumannii*. For this reason, we conceived the idea to create a synthetic report on this topic including updated epidemiology data, a description of CRE resistance patterns, current strategies of antimicrobial treatment, and future perspectives. From this update it is clear that antimicrobial stewardship and precision medicine are becoming essential to fight the emergence of antimicrobial resistance and that even if there are new drugs effective against CRE causing secondary peritonitis, these drugs have to be used carefully especially in empirical therapy.

## 1. Introduction

Primary, secondary, and tertiary peritonitis can be distinguished among the large class of intra-abdominal infections (IAIs). Primary peritonitis is defined as peritoneal inflammation and infection caused by bacterial translocation, blood, or iatrogenic spread without macroscopically evident lesion of solid organs or viscera of the gastrointestinal or genitourinary tract. Spontaneous bacterial peritonitis is attributable to primary peritonitis. Tertiary peritonitis is the natural evolution of secondary peritonitis with treatment failure (persistence of peritonitis 48 h after treatment) [1]. Secondary peritonitis, the main topic of this review, is among the major causes of community-acquired sepsis. It can be defined as the direct contamination of the peritoneum due to an injury (perforation) to a hollow viscus in the abdominal cavity (i.e., gastrointestinal tract or genitourinary tract). This lesion can be spontaneous (i.e., spontaneous ulceration/neoplastic expansion), or iatrogenic in the case of a postsurgical perforation (i.e., loss from an anastomosis). It is, therefore, to be noted that secondary peritonitis can frequently be due to complications of the surgical technique and related to exposure to polymicrobial contaminating sources. Contamination rarely comes from the injury of solid abdominal viscera or by suppurative abscesses. For the resolution of secondary peritonitis, it is necessary that surgical cleaning and washing of the infection site begin promptly and are associated with adequate and effective antibiotic treatment to prevent the development of tertiary peritonitis and sepsis [2]. It has to be taken into account that in recent years surgical management procedures have evolved compared to the 1990s. In fact, it is expected that for a patient with signs and symptoms of disseminated peritonitis (i.e., tense abdomen, not treatable) or sepsis (delta of SOFA score>2) an early treatment should be provided once his or her vital functions have stabilized, with a radical surgical intervention associated with an aggressive empirical antibiotic therapy first and then modulated on the antibiogram obtained from the samples of peritoneal fluid or purulent material taken during the surgery [3]. Most of the patients in this category are often already hospitalized in the ICU or will be managed in the ICU after surgery. In case of little signs and symptoms of peritonitis (i.e., localized and limited to a single abdominal quadrant or two at most) with laboratory and instrumental tests that confirm the circumscription of the inflammatory and of the infectious process, it is advisable to proceed with a less invasive intervention such as surgical cleaning and drainage associated with targeted antibiotic therapy after an antibiogram. Sometimes, only in highly selected (mild) cases, empirical antibiotic therapy can be evaluated as an early strategy alone before surgery [4]. In addition to the extension of the pathology in the context of secondary peritonitis, in order to estimate the severity of the clinical case, it is also recommended to classify the patient on the basis of comorbidities and predisposing factors for therapeutic failure [5]. Table 1 shows the predictive factors of failure in antimicrobial empiric treatment for secondary peritonitis.

Secondary peritonitis is among the most frequent causes of mortality (mortality rates 10–20%) in patients admitted to ICUs [6]. In some studies, such as that conducted in 2014 by De Waele et al., the death rate exceeds that of respiratory infections, especially in the case of multidrug-resistant pathogens. In particular, postsurgical secondary peritonitis, being caused by nosocomial pathogens, is the most frequent cause of septic shock and acute or multi-organ renal failure in already compromised patients. These are, therefore, postsurgical complications with extremely poor prognosis and difficult therapeutic management if caused by multidrug-resistant pathogens [7]. This comprehensive review of the literature aims to offer an overview of a very topical and complex management problem, secondary peritonitis, from multidrug-resistant pathogens with a focus on those caused by carbapenem-resistant *Enterobacterales*.

## 2. Epidemiology of Secondary Peritonitis Caused by Carbapenemase-Producing *Enterobacterales*

About 25% of cases of peritonitis are attributable to the secondary peritonitis group. Among the sites of origin of peritonitis in order of frequency, we find the colon, appendix, stomach/duodenum, small intestine, and biliary tract.

Two studies report data on the frequency of this pathology by stratifying patients according to the sites of origin of the infection. In 2009 Gauzit et al. reported the data in Figure 1.

Figure 2, on the other hand, describes the percentages of the perforation site by anatomical site.

It has to be noted that in 39% of cases the perforation originates from the colorectal, in 33% of cases from the stomach/duodenum, in 25% of cases from the small intestine, and in 3% of cases there is multiple perforation [6]. A second epidemiological study conducted in South Korea by Jang in 2015 reports, in agreement with what has already been reported, that secondary peritonitis derives mainly from the right colon (39.4% of cases). In 20% of cases it is associated with the positivity of the blood culture practiced on entry, and in more than 50% of cases it is associated with the positivity of the peritoneal fluid taken during surgical remediation. Secondary peritonitis is a clinical condition with high mortality (mortality rate 10–20%). This proportion increases up to 10% in the case of late or inadequate treatment [8]. The peritonitis with the worst prognosis is identified by a SAPS II (Simplified Acute Physiology II score) > 38, and usually this is peritonitis with non-appendiceal origin (Figure 3).

According to the findings of Gauzit et al. in 2009, the SAPS II had low values for peritonitis, starting with a better prognosis for peritonitis originating in the appendix and then increasing to high values for peritonitis originating in the colon [6]. Obviously, it would be reductive to correlate the prognosis only with the score obtained with the SAPS II. It must be considered that secondary peritonitis is caused by heterogeneous groups of pathogens such as Gram-negative bacteria, Gram-positive bacteria, and fungi, and cases of peritonitis caused by more than one pathogen at the same time have been reported. From this heterogeneity derives the difficulty in establishing an early empirical antibiotic treatment that is effective and reduces the possibility of MDR or XDR pathogens developing once the results of the antibiogram have been obtained and the clinical response of the patient to treatment has been assessed (approximately 48 h after sampling upon entrance) [8]. Focusing attention on carbapenemase-producing pathogens, as regards the year 2020, the European Center for Disease Prevention and Control reports the data on the incidence of infection by these pathogens. Table 2 summarizes the cases of infection described by pathogen and geographic location.

Figure 4 shows the geographical distribution of resistance to carbapenems expressed for each type of pathogen. These are overall data and do not only concern intra-abdominal infections.

It is evident that in Europe there is a wide diffusion of carbapenemase producer micro-organisms. Northern European countries have lower percentages of detection, while Italy, Greece, and Spain have higher percentages.

From these data it is clear that the problem of carbapenemase-producing infections goes well beyond one part of the body and that sometimes several parts are involved at the same time, making treatment very complex. Prevention, consisting of identifying risk factors and treating them early with appropriate therapeutic schemes, is the strategy that will enable a reduction in the incidence of these difficult-to-manage infections. A Spanish study highlighted the possibility of predicting the presence of multi-resistant contaminating pathogens, justifying an increase in aggression in the empirical treatment of these selected cases.

The risk factors for the development of nosocomial infections from multidrug-resistant microorganisms are summarized in Figure 5.

Patients with the risk factors shown in Figure 5 or patients with sepsis are early candidates for empirical treatment based on an antimicrobial drug with a broad spectrum of action. In these cases, in the absence of early antibiotic aggression of the pathogen, there is a high probability that the patient has a bad prognosis and, for this reason, the empirical use of a combination of latest-generation broad-spectrum antibiotics is justified [9].

As described in Figure 5, there are many predisposing factors for colonization by carbapenemase-resistant *Enterobacterales*: previous hospitalization in the ICU, previous long-term hospitalization, COPD, previous treatment with broad-spectrum antibiotics (beta-lactams + beta-lactamase inhibitors or carbapenems), previous MDR infection, and quinolone prophylaxis. Quinolone prophylaxis seems to play an important role in the emergence of GNB antimicrobial resistance. In fact, 70% of isolates with quinolone resistance show trimethoprim-sulfamethoxazole resistance.

Using these predisposing factors, we are able to identify the patients most predisposed to the onset of infections by resistant bacteria.

To effectively manage these infections, however, we must be aware of the characteristics expressed by each pathogen and the pathogen’s resistance profiles in order to adapt the treatment as much as possible, maximizing effectiveness and reducing doses and future resistance.

## 3. Resistance Patterns of *Enterobacterales* Causing Secondary Peritonitis

Antibiotic resistance is defined as the ability developed by some pathogens to inhibit the antimicrobial properties of the antibiotic classes used in the clinical setting and previously effective in limiting bacterial growth. It is a set of heterogeneous antimicrobial inactivation mechanisms which become increasingly complex to counteract. Currently, in fact, the development of new antimicrobial molecules effective against multidrug-resistant pathogens fails to keep pace with the rapid development of mechanisms of inactivation and evasion of pathogens that cause nosocomial infections. As regards secondary peritonitis, these are more frequently caused in the context of Gram-positive pathogens by methicillin-resistant Staphylococcus aureus (MRSA) or vancomycin-resistant Enterococcus faecium, and in the case of Gram-negative bacteria, non-fermenting pathogens such as *P. aeruginosa* and *A. baumannii* are highlighted. *A. baumannii*, *B. fragilis*, and *Enterobacterales* such as *E. coli* and *K. pneumoniae* produce β-lactamases (ESBLs and/or AmpC and/or carbapenemases) [10]. Therefore, given the increase in resistance both to vancomycin and to fluoroquinolones and aminoglycosides first, followed then by resistance to carbapenems, there is also difficulty in the use of these molecules which until now had been recognized as the last effective strategy of treatment. In summary, resistance to carbapenems is due to a combination of alterations in permeability and production of carbapenemases (non-metallo-carbapenemases, OXA-beta-lactamases, or IMP/VIM-beta-lactamases). Carbapenemases are β-lactamases capable of recognizing almost all hydrolysable beta-lactam antibiotics and of inactivating them. Carbapenemases can be classified according to the Amber Classification into four categories (A, B, C, and D). Class B carbapenemases are characterized by the presence of an active site containing zinc, while the other classes present serine at the active site [11]. The carbapenemases most represented in *Enterobacterales* are of categories A, B, and D. KPC and GES belong to class A, MBLs (IMP and VIM) belong to class B, and oxacillinases (OXA) belong to class D. KPC effectively hydrolyzes carbapenems, cephalosporins, penicillins, and aztreonam and is effectively inactivated by either clavulanic acid or tazobactam. IMP and VIM are similar in hydrolytic activity to class A carbapenemases, but unlike them do not hydrolyze aztreonam [12].

Because zinc-chelating EDTA possesses activity linked to the activation of the galvanized domain of the active site, its use could be helpful in reducing the activity of these beta-lactamases [13]. Type D carbapenemases are capable of hydrolyzing oxacillin and cloxacillin, and given the extreme variability of the active site (small and hydrophobic), they are poorly inhibited by clavulanic acid or EDTA [14]. It should be noted that the active site of these oxacillinases is sensitive to CO_2_ and is activated by high concentrations of CO_2_ [15]. In addition to the production of carbapenemases, multidrug-resistant pathogens are capable of inactivating carbapenems by associating the production of broad-spectrum beta-lactamases such as ESBLs and AmpCs with modified porins [16]. The last but not negligible mechanism of resistance is represented by the expression of efflux pumps and alterations in the penicillin-binding target proteins. 

## 4. The Emerging Problem of Carbapenem-Resistant *Enterobacterales* Causing Secondary Peritonitis

Among the *Enterobacterales* most frequently responsible for secondary peritonitis resistant to carbapenems, we find *K. pneumoniae* and *E. coli*, but other pathogens that do not belong to the class of *Enterobacterales* such as *P. aeruginosa* and *A. baumannii* cannot be neglected.

The worldwide spread, such as the European diffusion of carbapenem-resistant *Enterobacterales*, is mainly due to a selection of these pathogens over the years.

In fact, the increasingly frequent use, especially in monotherapy, of carbapenems to cope with infections caused by pathogens resistant to beta-lactams (penicillins and cephalosporins), fluoroquinolones, or aminoglycosides has made it possible for these pathogens to develop the ability to produce enzymes capable of degrading even carbapenems, making even one of our latest-generation molecules poorly effective. In Europe, from data reported in 2016, pathogens resistant also to carbapenems have already spread as endemic species [17].

In any case, we can schematize the resistance profiles to carbapenems based on the production by the pathogen of different types of carbapenemases [18].

Vancomycin-resistant enterococci deserve a separate treatment and will not be covered in detail in this review. For these pathogens, a study conducted by Wenstein has shown that in vitro resistance to penicillin, ampicillin, and vancomycin can predict resistance to imipenem in vivo. [19] In this case, the association between doxycycline linezolid and daptomycin is useful.

Table 3, Table 4, Table 5 and Table 6 summarize the sensitivity and resistance to different carbapenems commonly used in ICU for each of these carbapenemase-producing pathogens responsible for intra-abdominal infections. The data have been collected from the SENTRY database for the years 2019, 2020, and 2021. It was possible to extrapolate data on the resistance to carbapenems of each aforementioned pathogen.

## 5. Current Strategies of Antimicrobial Treatment of Secondary Peritonitis due to CRE

The increasingly frequent finding of multidrug-resistant pathogens is precisely the result of the abuse of broad-spectrum antibiotics. Thienamycin was discovered in 1978 from Streptomyces cattleya, and its carbapenems were synthesized in the laboratory in 1985. To date, these antibiotics have represented, for about thirty years, the most up-to-date treatment strategy for particularly aggressive pathogens [20]. 

The development of resistance to these antimicrobials and the non-circumscription of these pathogens in the hospital setting marked a new era in the treatment of nosocomial and community-based infections.

The strategies currently used to treat secondary peritonitis and other diseases caused by multidrug-resistant pathogens include early and effective surgical therapy in association with antibiotic therapy [21].

Preferably, an empirical combination antibiotic therapy should be evaluated in order to exploit the synergy between antibiotics, reduce their dosage, and at the same time decrease the mutagenic stress to which the bacteria causing the infection are subjected until data are available to undertake a targeted therapy [22]. 

At the basis of correct and early treatment, there is, therefore, the need to use an empirical antibiotic therapy that avoids exacerbating the development of antibiotic resistance especially using an exaggerate and unnecessary dosage of drugs [23]. However, at present there are no studies that define a clear superiority of combination therapy over monotherapy; it is therefore good to evaluate the epidemiology of the region of onset of the disease. Infections caused by CRE will usually be poorly responsive to treatment with penicillins, cephalosporins (with or without classical beta-lactamase inhibitors), and carbapenems; the sensitivity to aminoglycosides (gentamicin and amikacin), aztreonam, and trimethoprim/sulfamethoxazole is scarce or unpredictable. Once the possibility of a multidrug-resistant pathogen has been identified or resistance to carbapenems confirmed, our treatment options include the use of antibiotics alone or in combination. Although the therapeutic options seem multiple, it is difficult to use the correct antibiotic effectively while minimizing adverse events [24]. Below is a brief description of the antibiotics effective on CRE, visually summarized in Figure 6. 

As shown already in 2011 by Falagas and colleagues, tigecycline with colistin (sinergistic action), colistin with a carbapenem (sinergistic action), fosfomycin with a carbapenem (additive or synergistic action), fosfomycin with an aminoglycoside (additive or synergistic action), and a carbapenem with an aminoglycoside (synergistic action) have been reported as antibiotic combinations effectively administered to series of patients infected with carbapenemase-producing *Enterobacterales* [25]. 

Colistin or Polymyxin E (PXE) is a molecule still used today in monotherapy or in combination therapy with other drugs. It is very active on non-fermenting pathogens; however, new findings are showing that it is becoming poorly effective against KPC producers [26]. It has a narrow therapeutic window (an average concentration at steady state of 2 μg/mL is needed to achieve therapeutic targets, except where a concentration of approximately 2.5 μg/mL results in renal toxicity), which makes it unmanageable to use alone. A reduction in nephrotoxicity has been documented when associated with ascorbic acid [27]. 

Polymyxin B (PXB) is less nephrotoxic than PXE and is also available on adsorption filter (Toraymyxin) for patients suffering from endotoxin-mediated septic shock unresponsive to conventional therapy. Already, in a randomized study conducted in 2015 by Payen et al., it was reported that Polymyxin B fails to give advantages if used early in patients with secondary peritonitis and septic shock in the initial phase. Mortality and organ failure could be assessed if used after hemodynamic stabilization of these septic patients [28]. 

Fosfomycin (FOF) is a first-generation drug that is a valid aid in multi-antibiotic therapy (i.e., with aminoglycosides) in the management of both Gram-negative and Gram-positive infections. Its use as a single antimicrobial is not recommended given the proven speed of CRE in developing resistance [29].

Plazomicin (PLZ) is a new-generation aminoglycoside with reduced side effects and MIC values lower than those of amikacin. PLZ received approval for the treatment of complicated urinary tract infections in patients who are not candidates for other treatments [30]. It is active against Gram-negative producers of ESβL, KPC, and AmpC but not effective against MβL producers and poorly effective against non-fermenting pathogens [31].

High-dose tigecycline (TG) is used as a last resort. Being a glycylcycline, it is a valuable aid in the treatment of both Gram-negative and Gram-positive infections including *Pseudomonas* spp. and *Proteus* spp. Resistance to TG in multitherapy (i.e., with aminoglycosides) is unlikely to develop, but should it occur for the production of efflux pumps or ribosomal protection mechanisms, an intervention using everacycline is still possible [32].

Eravacycline (EV) is a fully synthetic fluorocyclin and an alternative to tigecycline in cases of resistance. It has good tolerability and a good safety profile. A further advantage is its excellent oral bioavailability, which facilitates administration even at home. It is approved for complicated intra-abdominal infections caused by pathogens producing ESβL, KPC, AmpC, MβL, and OXA, but it lacks activity against P.aeruginosa [33].

Cefiderocol (CFD) is a novel cephalosporin with affinity mainly for the penicillin-binding protein 3 (PBP3) of *Enterobacterales* and non-fermenting bacteria. It has shown a characteristic antibacterial spectrum with potent activity against a broad range of aerobic GNB, including carbapenem-resistant strains of *Enterobacterales* and non-fermenting pathogens such as *A. baumannii* and *P. aeruginosa*; it should be a valid alternative to peritonitis caused by MDR pathogens. Due to its chemical structure, CFD possesses stability to hydrolysis by almost all β-lactamases, including serine- and metallo-β-lactamases. CFD is considered a valuable resource for the treatment of patients with infections due to aerobic Gram-negative carbapenemase-producing bacteria with limited therapeutic options [34].

Ceftolozane/tazobactam (CTZ, with metronidazole M): The association ceftolozane/tazobactam can be considered as an efficient and safe carbapenem-sparing alternative for treating intra-abdominal infections [35].

Ceftazidime/avibactam (CAZ) is a combination of third-generation cephalosporin ceftazidime and avibactam, a non-beta-lactam beta-lactamase inhibitor, which restores the activity of ceftazidime against many beta-lactamase–producing GNB. Ceftazidime-avibactam is safe and has high tolerability; it produces clinical cure rates comparable to meropenem and is suggested for indications of complicated intra-abdominal infections (in association with metronidazole) [36].

Aztreonam/avibactam (AZAV): The AZAV combination has been shown to have in vitro activity against metallo-β-lactamase–producing bacteria, such as NDM/VIM/IMP. AZAV is the only β-lactam insensitive to the hydrolysis of metallo-β-lactamases. Unfortunately, metallo-β-lactamase–producing strains are often coproducers of β-lactamases capable of degrading Aztreonam (i.e., AmpC). Avibactam is able to inhibit β-lactamases which degrade Aztreonam, making this combination an effective tool in the treatment of infections caused by class B carbapenemase-producing strains. The association has been promoted to offer an excellent therapeutic opportunity against metallo-β-lactamase–producing strains [37,38].

Meropenem/vaborbactam (MV) in monotherapy was associated with increased clinical cure, decreased mortality, and reduced nephrotoxicity. Vaborbactam is a non-beta-lactam beta-lactamase inhibitor with a structure unlike any other currently marketed beta-lactamase inhibitor. Meropenem acquires more effectiveness by adding vaborbactam against most species of *Enterobacterales*. In vitro and in vivo pharmacodynamic studies have reported bactericidal activity against various Gram-negative strains, including carbapenem-resistant strains, with the exception of bacteria-producing MBL [39].

The relebactam + imipenem-cilastatin (RIC) association uses relebactam, a parenteral, small-molecule beta-lactamase inhibitor that is active against beta-lactamases; in vitro susceptibility studies have demonstrated that relebactam restores imipenem susceptibility to many imipenem-resistant bacteria. In vivo infection models show that relebactam given with imipenem-cilastatin could be used to treat severe Gram-negative infections [40,41].

Having briefly described the antibiotics currently used to treat carbapenem-resistant *Enterobacterales* infections, we outline possible therapeutic strategies in Figure 7.

## 6. Future Perspectives

Future perspectives on AMR management are currently focused on the advice of precision medicine and artificial intelligence. According to WHO data collected in 2020, more patients are expected to die from infections with MDR pathogens by 2050 than from cancer today, only 50% of antibiotics are used correctly globally, and drug-resistant infections cause at least 700,000 deaths worldwide per year (projected to rise to 10 million deaths per year by 2050). Therefore, corrective measures have to be taken using any means [42].

Compelling evidence has demonstrated the effectiveness of antibiotic stewardship programs in reducing inappropriate antimicrobial use, AMR rates, hospital-acquired AMR, hospitalization time, and costs. Despite these successes, ongoing challenges for antibiotic stewardship programs include rapid differentiation between bacterial and viral infections, early pathogen identification and characterization (i.e., antibiotic susceptibility), and reduction in antimicrobial abuse overwhelmed during the COVID-19 pandemic. Approximately 75% of the patients diagnosed with COVID-19 received antibiotics, increasing the risk for acquiring AMR infections. Hence the necessity to improve antimicrobial management, supplementing antibiotic stewardship with precision medicine.

Precision medicine (PM) is an innovative multicomponent medical approach designed to optimize efficiency and advantages for particular groups of patients using genetic or molecular profiling. More practically, precision medicine involves rapidly identifying altered biology within a patient and using the findings to guide therapy.

Precision medicine was first developed in the field of oncology. Its use in infectious diseases is at an early stage, but it is showing great potential, especially in employing “omics”-based biomarkers such as proteomics, metabolomics, and lipidomics to estimate disease prognosis, predict treatment response, and improve clinical outcomes [43].

In the case of abdominal infections and sepsis, some of these approaches are still in the early stages of research, while others, even if they are not specific markers (i.e., biomarkers), are already in routine use in clinical practice [44].

Biomarkers such as procalcitonin (PCT) and the mid-region fragment of pro-adrenomedullin (MR-proADM) may be less specific but useful drivers to properly detect the efficacy of the ongoing therapeutic approach. Dosing these molecules several times during hospitalization is expensive, so it is fundamental to identify subpopulations who could benefit from this marker’s detection to improve a poor prognosis.

The use of PCT-guided algorithms for antimicrobial stewardship in sepsis has shown a reduction in mortality. A value of PCT > 2 ng/mL seems to identify patients that may benefit from receiving adjuvant therapy with hydrocortisone, vitamin C, and thiamine to reduce the progression of organ dysfunction in case of septic shock, while in patients with high initial levels of PCT a more aggressive treatment can be justified because levels of PCT > 6 ng/mL predict progressive organ dysfunction and increased risk of mortality [45]. 

Interestingly, PCT repetitive dosage can be used as a predictor of adverse outcomes and treatment failure because the non-clearance of PCT by more than 80% is a significant independent predictor of mortality [46].

The other biomarker, produced by vascular endothelial cells, is the MR-proADM, which directly reflects plasma levels of adrenomedullin (a vasodilator agent with metabolic and immune-modulating properties). MR-proADM if detected early in plasma reflects a worse prognosis, predicting mortality better than lactate rise and SOFA score, ICU admission, and the need for urgent treatment. High-plasma MR-proADM clearance at day 5 is related to better outcomes [44,47]. 

Furthermore, artificial intelligence and machine learning algorithms hold great promise for integrating precision medicine approaches with antibiotic stewardship. There are three main areas in which investigators are currently focused: (a) antimicrobial resistance prediction in interpreting genomic data; (b) deepening the cellular functions disrupted by antibiotics and developing novel antimicrobial agents; and (c) taking antimicrobial stewardship decisions using data extracted from electronic medical recorders [48]. 

A huge amount of healthcare information (e.g., signs and symptoms, genetics, risk factors, immune response, molecular enzymes, virulence factors, environment, and epidemiology) can be managed by artificial intelligence and used to deliver the most efficient treatment or preventive care in a timely manner, thus minimizing risks of mistakes and adverse events. 

In the near future, both precision medicine and artificial intelligence will a play a leading role in antibiotic stewardship, especially in ICU-complicated infections. 

## 7. Search Strategy

Several internet databases were consulted (Ovid MEDLINE, EMBASE, and Google Scholar), using a combination of terms such as “carbapenem resistance,” “secondary peritonitis,” “severe infections,” “intensive care unit,” “multi-resistant bacteria,” and “*Enterobacterales*” to identify studies (without date limitations) reporting secondary peritonitis due to carbapenem-resistant *Enterobacterales*. The resulting studies were listed using a reference manager software (Endnote 20. Clarivate Analytics, Philadelphia, PA, USA), and duplicates were removed. Results and primary evidence of the studies thus obtained were summarized in narrative form and through figures and tables.

The SENTRY Public Dataset by JMI laboratories and the European Centre for Disease Prevention and Control was consulted to find the resistance patterns of all infections and intra-abdominal infections. The data obtained are summarized in tables and discussed in the epidemiology section.

## Figures and Tables

**Figure 1 antibiotics-11-01347-f001:**
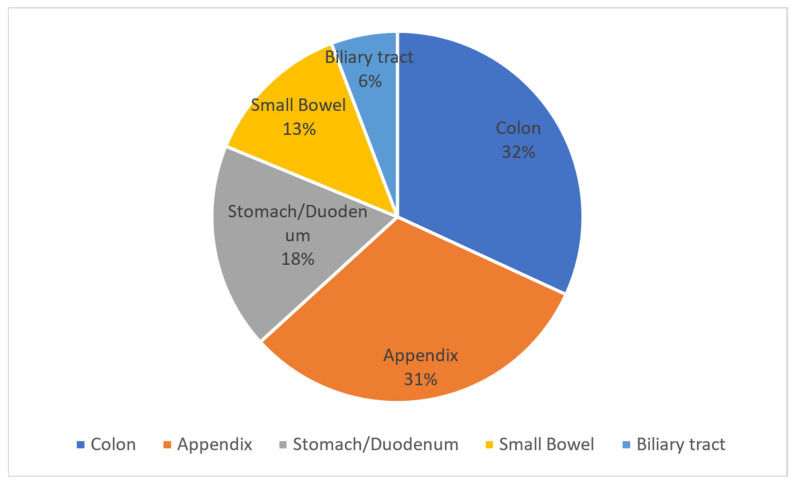
This cake diagram illustrates the percentage of sites of infections causing secondary peritonitis according to the viscus of origin.

**Figure 2 antibiotics-11-01347-f002:**
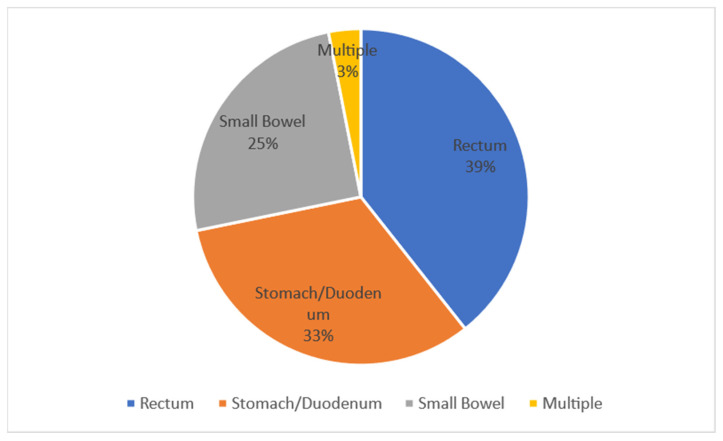
This pie chart illustrates the percentages of the perforation site by anatomical site.

**Figure 3 antibiotics-11-01347-f003:**
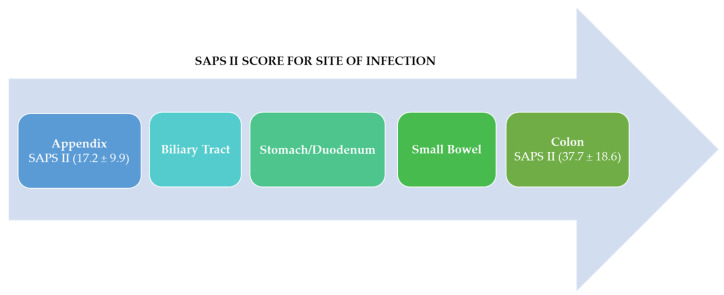
This arrow diagram represents SAPS II from higher values to lower values. The SAPS II was higher for the colon site of infection ranging 37.7 ± 18.6 than for the appendix ranging 17.2 ± 9.9.

**Figure 4 antibiotics-11-01347-f004:**
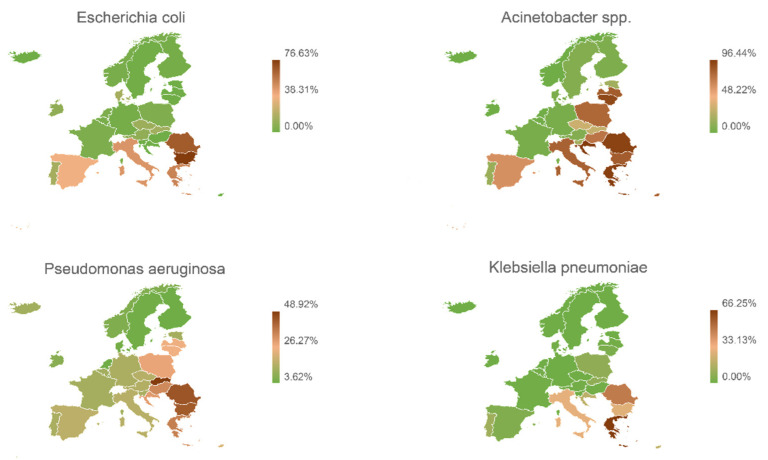
Geographical European distribution of the resistance to carbapenems expressed for each type of pathogen.

**Figure 5 antibiotics-11-01347-f005:**
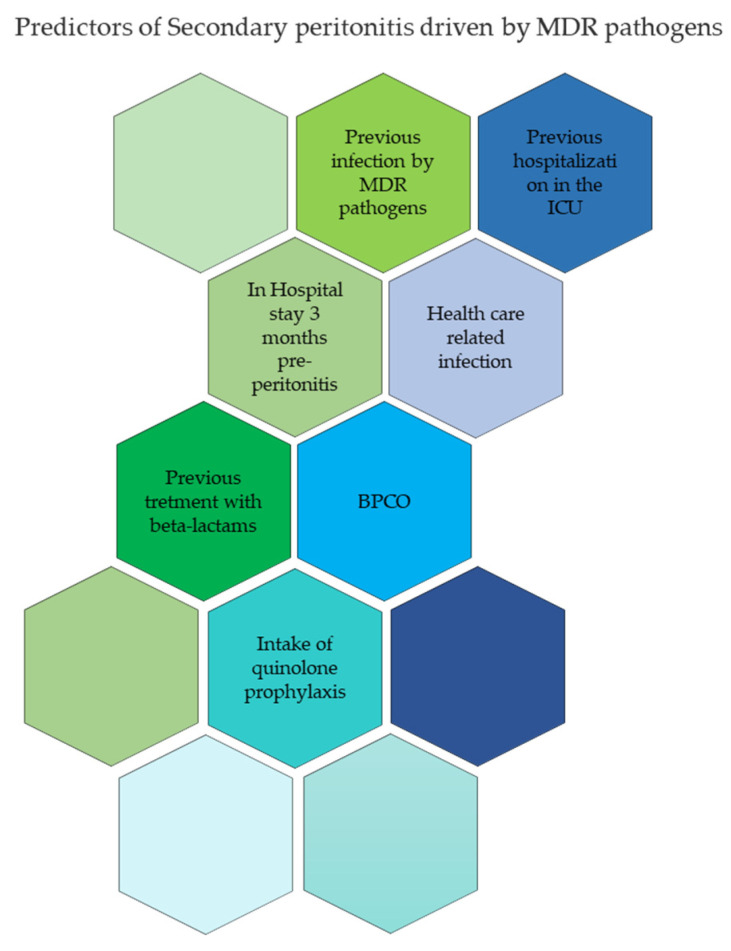
Graphic synthesis of the predictive factors of MDR-driven peritonitis.

**Figure 6 antibiotics-11-01347-f006:**
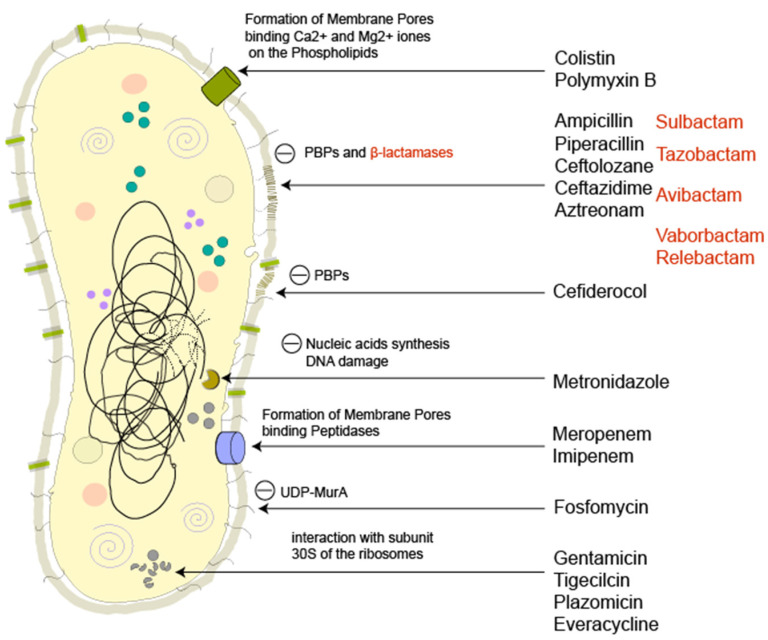
Illustration of a Gram-negative bacteria and the site of action of all the antimicrobial molecules cited in this comprehensive review. The figure is an original production by S. Di Franco and A. Alfieri. Notes — PBPs: penicillin-binding proteins; UDP-MurA: UDP-N-acetylmuramic acid.

**Figure 7 antibiotics-11-01347-f007:**
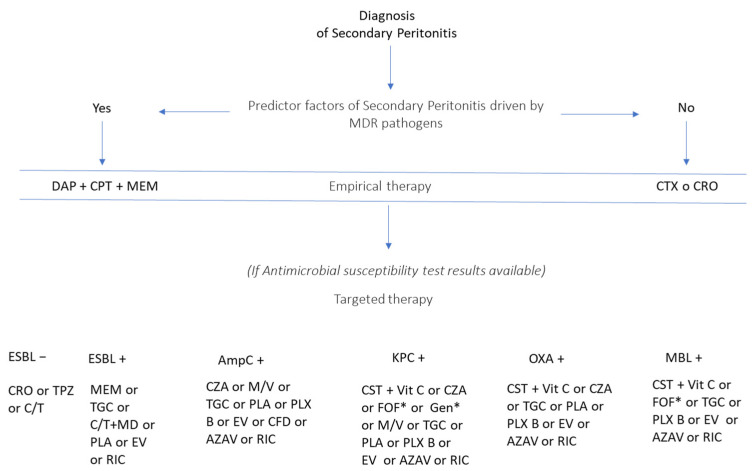
Pharmacological options to treat secondary peritonitis by carbapenem-resistant pathogens. Notes: AZAV-aztreonam/avibactam; CAZ- ceftazidime/avibactam; CFD-cefideclor; CPT-ceftaroline; CRO-ceftriaxone; CT- ceftolozane/tazobactam; CTX-cefotaxime; DAP-daptomicine; EV-everacicline; FOF-fosfomicine; GEN-gentamicin; MD-metronidazole; MEM-meropenem; MV-meropenem/vaborbactam; PLA-plazomycin; PXB-polymyxin B; PXE-polymyxin E; RIC-relebactam/imipenem/cilastatin; TG-tigecycline; TPZ-piperacillin/tazobactam; VitC-ascorbic acid. *not in monotherapy.

**Table 1 antibiotics-11-01347-t001:** Predictive factors of failure in source control for secondary peritonitis.

Predictive Factors of Failure in Source Control for Secondary Peritonitis
Delay in the initial intervention (>24 h)
High severity of illness (APACHE II * score ≥ 15; SOFA ** > 2)
Advanced age, Degree of peritoneal involvement (>2 abdominal quadrants), or diffuse peritonitis
Inability to achieve adequate debridement or control by surgical toilette/drainage
Advanced age (>56 y.o.) and gender (males > females)
Poor nutritional status
Low albumin level
Comorbidities and organ dysfunction
Presence of malignancy

* APACHE, Acute Physiology and Chronic Health Evaluation; ** SOFA score, Sequential Organ Failure Assessment Score.

**Table 2 antibiotics-11-01347-t002:** Report on the European percentage of carbapenemase-producing pathogens from the European Center for Disease Prevention and Control. In the table the red color gives a visual representation of the percentage of resistance in each country.

European Centre for Disease Prevention and Control							
percentage of isolates resistant to carbapenems in europe 2020 (all diseases)							
*Klebsiella pneumoniae*		*Acinetobacter* spp.		*Escherichia coli*		*Pseudomonas aeruginosa*	
Greece	66.25%	Croatia	96.44%	Bulgaria	76.63%	Slovakia	48.92%
Romania	48.31%	Greece	94.59%	Romania	66.08%	Romania	43.92%
Italy	29.51%	Romania	93.27%	Greece	53.00%	Bulgaria	42.86%
Bulgaria	28.11%	Lithuania	91.08%	Italy	47.22%	Greece	35.71%
Cyprus	19.77%	Bulgaria	82.95%	Spain	37.19%	Hungary	33.76%
Croatia	19.10%	Latvia	82.69%	Portugal	15.43%	Croatia	30.30%
Portugal	11.58%	Cyprus	81.03%	Denmark	15.41%	Poland	28.48%
Poland	8.19%	Italy	80.80%	Slovakia	14.18%	Lithuania	25.62%
Slovakia	8.16%	Poland	78.23%	Czechia	13.33%	Latvia	25.58%
Malta	7.58%	Hungary	73.03%	Austria	9.73%	Cyprus	20.63%
Spain	4.72%	Spain	61.54%	Ireland	9.50%	Spain	16.60%
Lithuania	2.91%	Czechia	32.93%	Poland	4.81%	Italy	15.95%
Luxembourg	1.15%	Slovakia	30.77%	Netherlands	4.01%	Czechia	15.74%
Belgium	1.14%	Slovenia	19.44%	France	3.92%	Austria	15.08%
Latvia	1.06%	Estonia	18.18%	Norway	2.74%	Germany	13.80%
Austria	0.95%	Portugal	15.38%	Belgium	2.42%	Slovenia	13.44%
Denmark	0.78%	Austria	7.25%	Finland	1.86%	Portugal	13.43%
Hungary	0.69%	Sweden	7.14%	Germany	1.82%	Estonia	12.66%
France	0.54%	Finland	5.41%	Sweden	1.02%	France	12.64%
Czechia	0.49%	Denmark	4.69%	Cyprus	0.00%	Belgium	12.45%
Germany	0.45%	Germany	3.46%	Estonia	0.00%	Iceland	12.00%
Sweden	0.27%	France	3.32%	Croatia	0.00%	Luxembourg	8.51%
Ireland	0.27%	Belgium	1.25%	Hungary	0.00%	Malta	8.16%
Norway	0.15%	Netherlands	0.67%	Iceland	0.00%	Ireland	7.77%
Finland	0.11%	Ireland	0.00%	Lithuania	0.00%	Norway	6.38%
Netherlands	0.07%	Norway	0.00%	Luxembourg	0.00%	Denmark	4.37%
Estonia	0.00%	Iceland	0.00%	Latvia	0.00%	Sweden	4.23%
Iceland	0.00%	Luxembourg	0.00%	Malta	0.00%	Finland	3.70%
Slovenia	0.00%	Malta	0.00%	Slovenia	0.00%	Netherlands	3.62%
Mean	9.04%	Mean	36.74%	Mean	13.11%	Mean	18.49%
Min	0.00%	Min	0.00%	Min	0.00%	Min	3.62%
Max	66.25%	Max	96.44%	Max	76.63%	Max	48.92%

**Table 3 antibiotics-11-01347-t003:** Activity of antimicrobial agents tested against 82 *Acinetobacter* isolates from intra-abdominal infections.

Activity of antimicrobial agents tested against 82 ***Acinetobacter*** isolates from intra-abdominal infections in the SENTRY program collected during 2019, 2020, and 2021											
Organisms include: *Acinetobacter baumannii-calcoaceticus* species complex (70), *A. haemolyticus* (1), *A. johnsonii* (1), *A. proteolyticus* (2), *A. radioresistens* (1), *A. schindleri* (2), *A. soli* (2), *A. ursingii* (3)											
**Antimicrobial Agent**	**Continent**	**Count**	**MIC50**	**MIC90**	**Range**	**CLSI**			**EUCAST**		
						**S (%)**	**I (%)**	**R (%)**	**S (%)**	**I (%)**	**R (%)**
**Imipenem**	**All**	82	0.25	>8	≤0.12 to >8	50.0	0.0	50.0	50.0	0.0	50.0
	Asia-W. Pacific	24	0.25	>8	≤0.12 to >8	50.0	0.0	50.0	50.0	0.0	50.0
	Europe	26	0.25	>8	≤0.12 to >8	50.0	0.0	50.0	50.0	0.0	50.0
	Latin America	17	>8	>8	≤0.12 to >8	17.6	0.0	82.4	17.6	0.0	82.4
	North America	15	0.25	>8	≤0.12 to >8	86.7	0.0	13.3	86.7	0.0	13.3
**Meropenem**	**All**	82	2	>32	0.12 to >32	50.0	0.0	50.0	50.0	0.0	50.0
	Asia-W. Pacific	24	0.5	>32	0.12 to >32	50.0	0.0	50.0	50.0	0.0	50.0
	Europe	26	1	>32	0.12 to >32	50.0	0.0	50.0	50.0	0.0	50.0
	Latin America	17	>32	>32	0.12 to >32	17.6	0.0	82.4	17.6	0.0	82.4
	North America	15	0.5	>32	0.12 to >32	86.7	0.0	13.3	86.7	0.0	13.3

Abbreviations (MIC50 and MIC90: the lowest concentration of the antibiotic at which 50 and 90% of the isolates were inhibited; CLSI: Clinical and Laboratory Standards Institute; EUCAST: European Committee on Antimicrobial Susceptibility Testing; S (%) percentage of susceptible pathogens, evidenced in green; I (%) percentace of pathogens with intermediate susceptibility, evidenced in yellow; R (%) percentage of resistant pathogens, evidenced in red).

**Table 4 antibiotics-11-01347-t004:** Activity of antimicrobial agents tested against 944 *Klebsiella* isolates from intra-abdominal infections.

Activity of antimicrobial agents tested against 944 ***Klebsiella*** isolates from intra-abdominal infections in the SENTRY program collected during 2019, 2020, and 2021											
Organisms include: *Klebsiella aerogenes* (69), *K. oxytoca* (153), *K. pneumoniae* (693), *K. variicola* (29)											
**Antimicrobial Agent**	**Continent**	**Count**	**MIC50**	**MIC90**	**Range**	**CLSI**			**EUCAST**		
						**S (%)**	**I (%)**	**R (%)**	**S (%)**	**I (%)**	**R (%)**
**Ertapenem**	**All**	709	0.015	0.5	≤0.008 to >2	90.6	0.8	8.6	90.6	0	9.4
	Asia-W. Pacific	150	≤0.008	>2	≤0.008 to >2	88.0	1.3	10.7	88.0	0	12.0
	Europe	208	0.015	>2	≤0.008 to >2	83.2	0.5	16.3	83.2	0	16.8
	Latin America	56	0.015	1	≤0.008 to >2	89.3	1.8	8.9	89.3	0	10.7
	North America	295	0.015	0.12	≤0.008 to >2	97.3	0.7	2.0	97.3	0	2.7
**Imipenem**	**All**	944	≤0.12	1	≤0.12 to >8	93.0	0.5	6.5	93.5	0.7	5.7
	Asia-W. Pacific	213	≤0.12	1	≤0.12 to >8	92.5	0.5	7.0	93.0	0.0	7.0
	Europe	324	≤0.12	2	≤0.12 to >8	88.9	1.2	9.9	90.1	1.5	8.3
	Latin America	79	≤0.12	8	≤0.12 to >8	88.6	0.0	11.4	88.6	1.3	10.1
	North America	328	≤0.12	0.5	≤0.12 to >8	98.5	0.0	1.5	98.5	0.3	1.2
**Meropenem**	**All**	944	0.03	0.12	≤0.015 to >32	93.1	0.4	6.5	93.5	1.6	4.9
	Asia-W. Pacific	213	0.03	0.12	≤0.015 to >32	92.5	0.0	7.5	92.5	1.9	5.6
	Europe	324	0.03	4	≤0.015 to >32	89.2	0.6	10.2	89.8	2.8	7.4
	Latin America	79	0.03	4	≤0.015 to >32	88.6	1.3	10.1	89.9	1.3	8.9
	North America	328	0.03	0.03	≤0.015 to >32	98.5	0.3	1.2	98.8	0.3	0.9

Abbreviations (MIC50 and MIC90: the lowest concentration of the antibiotic at which 50 and 90% of the isolates were inhibited; CLSI: Clinical and Laboratory Standards Institute; EUCAST: European Committee on Antimicrobial Susceptibility Testing; S (%) percentage of susceptible pathogens, evidenced in green; I (%) percentace of pathogens with intermediate susceptibility, evidenced in yellow; R (%) percentage of resistant pathogens, evidenced in red).

**Table 5 antibiotics-11-01347-t005:** Activity of antimicrobial agents tested against 1922 *Escherichia* isolates from intra-abdominal infections.

Activity of antimicrobial agents tested against 1922 *Escherichia* isolates from intra-abdominal infections in the SENTRY program collected during 2019, 2020, and 2021											
Organisms include: *Escherichia coli* (1921), *E. marmotae* (1)											
**Antimicrobial Agent**	**Continent**	**Count**	**MIC50**	**MIC90**	**Range**	**CLSI**			**EUCAST**		
						**S (%)**	**I (%)**	**R (%)**	**S (%)**	**I (%)**	**R (%)**
**Ertapenem**	**All**	1459	≤0.008	0.03	≤0.008 to >2	98.5	0.5	1.0	98.5	0	1.5
	Asia-W. Pacific	209	≤0.008	0.06	≤0.008 to >2	96.7	1.0	2.4	96.7	0	3.3
	Europe	510	≤0.008	0.03	≤0.008 to >2	99.0	0.8	0.2	99.0	0	1.0
	Latin America	178	≤0.008	0.06	≤0.008 to >2	99.4	0.0	0.6	99.4	0	0.6
	North America	562	≤0.008	0.03	≤0.008 to >2	98.4	0.4	1.2	98.4	0	1.6
**Imipenem**	**All**	1922	≤0.12	≤0.12	≤0.12 to >8	99.2	0.2	0.7	99.3	0.2	0.5
	Asia-W. Pacific	336	≤0.12	0.25	≤0.12 to >8	98.2	0.0	1.8	98.2	0.0	1.8
	Europe	775	≤0.12	≤0.12	≤0.12 to 4	99.6	0.3	0.1	99.9	0.1	0.0
	Latin America	249	≤0.12	0.25	≤0.12 to 8	98.4	0.0	1.6	98.4	0.8	0.8
	North America	562	≤0.12	≤0.12	≤0.12 to >8	99.5	0.2	0.4	99.6	0.0	0.4
**Meropenem**	**All**	1922	≤0.015	0.03	≤0.015 to >32	99.3	0.2	0.6	99.4	0.2	0.4
	Asia-W. Pacific	336	≤0.015	0.03	≤0.015 to >32	98.2	0.0	1.8	98.2	0.6	1.2
	Europe	775	≤0.015	0.03	≤0.015 to 16	99.9	0.0	0.1	99.9	0.0	0.1
	Latin America	249	≤0.015	0.03	≤0.015 to 32	98.4	0.8	0.8	99.2	0.0	0.8
	North America	562	≤0.015	0.03	≤0.015 to >32	99.5	0.2	0.4	99.6	0.2	0.2

Abbreviations (MIC50 and MIC90: the lowest concentration of the antibiotic at which 50 and 90% of the isolates were inhibited; CLSI: Clinical and Laboratory Standards Institute; EUCAST: European Committee on Antimicrobial Susceptibility Testing; S (%) percentage of susceptible pathogens, evidenced in green; I (%) percentace of pathogens with intermediate susceptibility, evidenced in yellow; R (%) percentage of resistant pathogens, evidenced in red).

**Table 6 antibiotics-11-01347-t006:** Activity of antimicrobial agents tested against 545 *Pseudomonas* isolates from intra-abdominal infections.

Activity of antimicrobial agents tested against 545 Pseudomonas isolates from intra-abdominal infections in the SENTRY program collected during 2019, 2020, and 2021											
Organisms include: *Pseudomonas aeruginosa* (529), *P. citronellolis* (1), *P. fluorescens* group (1), *P. koreensis* (1), *P. oryzihabitans* (1), *P. plecoglossicida* (1), *P. protegens* (2), *P. putida* group (4), *P. stutzeri* (3), unspeciated *Pseudomonas* (2)											
**Antimicrobial Agent**	**Continent**	**Count**	**MIC50**	**MIC90**	**Range**	**CLSI**			**EUCAST**		
						**S (%)**	**I (%)**	**R (%)**	**S (%)**	**I (%)**	**R (%)**
**Imipenem**	**All**	544	1	8	≤0.12 to >8	80.5	4.4	15.1	0	0	0
	Asia-W. Pacific	138	1	8	≤0.12 to >8	86.2	2.2	11.6	0	0	0
	Europe	184	1	8	≤0.12 to >8	76.1	7.6	16.3	0	0	0
	Latin America	52	1	>8	≤0.12 to >8	76.9	1.9	21.2	0	0	0
	North America	170	1	8	≤0.12 to >8	81.8	3.5	14.7	0	0	0
**Meropenem**	**All**	541	0.5	8	≤0.015 to >32	83.9	4.3	11.8	83.4	9.4	7.2
	Asia-W. Pacific	138	0.25	8	≤0.015 to >32	88.4	0.7	10.9	88.4	5.1	6.5
	Europe	182	0.5	8	≤0.015 to 32	81.3	6.6	12.1	81.3	12.6	6.0
	Latin America	50	0.5	16	0.03 to >32	78.0	6.0	16.0	78.0	10.0	12.0
	North America	171	0.5	8	0.03 to >32	84.8	4.1	11.1	83.0	9.4	7.6

Abbreviations (MIC50 and MIC90: the lowest concentration of the antibiotic at which 50 and 90% of the isolates were inhibited; CLSI: Clinical and Laboratory Standards Institute; EUCAST: European Committee on Antimicrobial Susceptibility Testing; S (%) percentage of susceptible pathogens, evidenced in green; I (%) percentace of pathogens with intermediate susceptibility, evidenced in yellow; R (%) percentage of resistant pathogens, evidenced in red).

## Data Availability

Not applicable.

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
