# Peer review of "A Literature Overview of Secondary Peritonitis Due to Carbapenem-Resistant Enterobacterales (CRE) in Intensive Care Unit (ICU) Patients"

_antibiotics, 2022, doi:10.3390/antibiotics11101347_

Round 1

Reviewer 1 Report

The main question addressed by this literature review is how MDR-pathogens can lead to secondary peritonitis in ICU. The topic is original as there are no other similar studies in the literature for the particular topic. This study will help physicians to prevent development of secondary peritonitis or treat it more efficiently. The methodology was sound and well-designed - no corrections. The conclusions are consistent with the evidence and arguments presented and the main question posed is well addressed. The references are appropriate and up-to-date. No additional comments on the tables and figures.

Minor corrections:

Line 22: Define CRE

Line 22-23: Correct and italicise microbial names

Abstract: What were your findings and conclusions? Why are they important?

Figures 1,2: Remove graph title

Line 284: Gram-negative

Figure 6: delete AA - SDF and move the date to the figure legend.

Titles 2,3: Provide more detail. Epidemiology, antibiotic resistance of what?

Italicise genus names throughout the text.

Section 7: were there any publication date filters?

References: correct the formatting inconsistencies, list references alphabetically.

Author Response

To Reviewer 1, thanks for your time. We have really appreciated your advices. 

As suggested we applied the following corrections:

- we have defined CRE in Line 22

- we have corrected and italicised microbial names in Line 22-23

- in the Abstract we have added findings and conclusionsù

- we have removed graph titles in Figures 1 and Figure 2

- we have correcter gram-negative in Line 284

- we have deleted AA - SDF from figure 6 and moved it the legend.

- we have added more details in Titles 2,3

- we have Italicised genus names throughout the text.

- in Section 7 we have clarified that there were not publication date filters

- we have corrected the formatting inconsistencies, list references alphabetically

Reviewer 2 Report

The authors present an interesting overview on secondary peritonitis complicated by MDR strains. However, the microbiological point of view needs to be strongly revised. So I recommend to correct the manuscript and add information according to below written recommendations.

I also recommend significantly improve graphic edit.

Regarding the title of the manuscript, I would recommend to remove abbreviations CRE and ICU. It is inappropriate. These abbreviations should be introduced in Abstract section.

Line 22 – Please correct the names of bacteria: P. aeruginosa, A. baumannii, it has to be also in italics, check it through the manuscript

Elsewhere: If you use “i.e.” you should italicize it and write comma behind it

Lines 88, 112, 289 – instead of colleagues use …et al. - elsewhere

Line 181 – MRSA is methicillin-resistant Staphylococcus aureus

Line 184 – Bacteroides fragilis does not belong among Enterobacteriaceae Family nor Enterobacterales order.

Otherwise E. coli and K. pneumoniae belongs to Family Enterobacteriaceae, but for example genus Serratia, Morganella belong into other Families, so you need to be careful and I recommend to use order Enterobacterales.

Line 191 – OXA-metallo-beta-lactamases is probably not correct term, since among metallo-beta-lactamases belong IMP, VIM, NDM  and they are in Ambler class B, whereas OXA are in group D (OXA-48) as it is mentioned below in manuscript.

Line 196 – I would recommend to use term Enterobacterales instead of Enterobacteriaceaie, also in title.

Line 201 – I miss in this section evaluation of inhibitors of carbapenemases….avibactam, relebactam, vaborbactam….

Line 217 – Enterobacteriaceae is Family, not class – again, I recommend to use Enterobacterales elsewhere

Line 230-233 – I would not claim that in vitro resistance to imipenem predicts resistance to vancomycin in vivo, since for example Enterococcus faecium is completely resistant to imipenem. Moreover, I miss the reference for whole paragraph.  

Line 284 – correctly it should be “gram-negative bacteria” – check it throughout the manuscript

Line 289-293 – what are the effects among these antibiotics’ combinations? Synergic, additive? Please add to the text.

Line 296 – there should be a reference for claim, that colistin can be poorly effective against KPC producers

Line 339 – ceftolozane/tazobactam

Line 362 – All carbapenem-resistant strains including those with MBL?

Author Response

To Reviewer 2, thanks for your time. We have really appreciated your advices. 

As suggested we applied the following corrections:

- we have corrected and italicised the names of bacteria (P. aeruginosaA. baumannii) in line 22 and through the manuscript

- we have italicised  “i.e.” through the manuscript

- we used et al. in Lines 88, 112, 289

- in Line 181 we have correcter MRSA 

- in Line 184 we have changed the sentence and we hope it is clear that Bacteroides fragilis does not belong among Enterobacteriaceae Family nor Enterobacterales order.

- we used the therm Enterobacterales through the manuscript and also in title.

- in Line 191 we have corrected OXA beta-lactamase

- the description of the action of inhibitors of carbapenemases (avibactam, relebactam, vaborbactam) is available in section 5. 

- Line 230-233 we have modified the text and added a reference

- we have corrected “gram-negative bacteria” throughout the manuscript

- in Line 289-293 we have added to the text the effects among the cited antibiotics’ combinations

- in Line 296 we have added a reference

- in Line 339 we have corrected ceftolozane/tazobactam

- in Line 362 we have changed the sentence to clarify the message

Round 2

Reviewer 2 Report

Regarding the microbiological part, I feel, that the manuscript may be accepted. Yet, I would recommend to remove abbreviations CRE and ICU from the manuscript topic.